# Senescence drives immunotherapy resistance by inducing an immunosuppressive tumor microenvironment

Damien Maggiorani[1,2], Oanh Le[1], Véronique Lisi[1], Séverine Landais [1], Gaël Moquin-Beaudry [3], Vincent Philippe Lavallée [1,4], Hélène Decaluwe [1,4,5] & Christian Beauséjour [1,2] ✉

The potential of immune checkpoint inhibitors (ICI) may be limited in situations where immune cell fitness is impaired. Here, we show that the efficacy of cancer immunotherapies is compromised by the accumulation of senescent cells in mice and in the context of therapy-induced senescence (TIS). Resistance to immunotherapy is associated with a decrease in the accumulation and activation of CD8 T cells within tumors. Elimination of senescent cells restores immune homeostasis within the tumor micro-environment (TME) and increases mice survival in response to immunotherapy. Using single-cell transcriptomic analysis, we observe that the injection of ABT263 (Navitoclax) reverses the exacerbated immunosuppressive profile of myeloid cells in the TME. Elimination of these myeloid cells also restores CD8 T cell proliferation in vitro and abrogates immunotherapy resistance in vivo. Overall, our study suggests that the use of senolytic drugs before ICI may constitute a pharmacological approach to improve the effectiveness of cancer immunotherapies.

Immunotherapies based on immune checkpoint inhibitors (ICI), by alleviating immunosuppressive pathways, constitute a game changer in cancer treatments[1,2]. However, depending on the cancer type, ~50–75% of treated patients are either resistant to immunotherapy or eventually relapse a few years after[3,4]. As a result, there is an urgent need to identify mechanisms that compromise the full potential of ICI-blockade therapy, especially in solid tumors.

We believe that the accumulation of senescent cells following exposure to genotoxic stresses, such as cancer treatments (radiotherapy and chemotherapy)[5–8], can induce immunotherapy resistance. Indeed, the accumulation of senescent cells is known to trigger age-related pathologies, and we recently showed that ionizing radiation-induced senescence greatly impairs immune cell functions in mice[9–12]. This phenotype could be reversed following the genetic elimination of

senescent cells[12]. Moreover, a recent study showed an increase in the number of senescent T cells with impaired proliferative capacity in human blood during aging[13]. Others showed that blocking DNA damage signaling can prevent tumor-specific T-cell senescence and enhance ICI efficacy against tumors[14]. Likewise, the use of spermidine in aged mice was shown to improve the efficacy of immunotherapy by restoring mitochondrial metabolism in CD8 T cells[15]. The accumulation of myeloid cells, especially myeloid-derived suppressor cells (MDSC), is an important mechanism that was shown to impair the tumor immune response[16,17]. Moreover, aging and exposure to radiotherapy were shown to favor myeloid skewing[18,19]. These observations lead us to hypothesize that senolytic drugs may be able to improve ICI based therapies by rejuvenating immune cell. Senolytic drugs were recently developed to specifically eliminate senescent cells which was shown to

[1]Centre de recherche du CHU Sainte-Justine, Montréal, QC, Canada. [2]Département de pharmacologie et physiologie (Université de Montréal, Montréal, QC, Canada. [3]Institut Gustave Roussy, Paris, France. [4]Département de pédiatrie (Université de Montréal, Montréal, QC, Canada. [5]Département de microbiologie, immunologie et infectiologie (Université de Montréal, Montréal, QC, Canada. ✉e-mail: c.beausejour@umontreal.ca

decrease the severity of the age-related pathologies[20,21]. In particular, ABT263 (Navitoclax), a potent BCL-2 family protein inhibitor was shown to efficiently remove senescent cells from lymphoid tissue and to decrease myeloid skewing in aged mice[18]. Finally, recent evidences have shown it is possible to clear senescent cells in human using senolytic drugs[22].

Here, using mouse carcinoma models, we show that αPD-L1 immunotherapy is compromised in mice previously exposed to cancer therapy and in the context of therapy-induced senescence (TIS). We also demonstrate that senescence impairs the abscopal effect, defined by the immune rejection of a secondary tumor distant from a primary tumor treated by radiotherapy[23,24]. Mechanistically, we show that both of these deleterious effects are associated with impaired CD8 T cell activation mediated by an immunosuppressive TME and are reversible upon treatment of mice with ABT263 or depletion of Ly6c+ myeloid cells. Our results suggest that the combination of a senolytic drug with ICI may be a strategy to improve the efficacy of immunotherapies.

## Results

### Senescence leads to immunotherapy resistance in mice

Senescent cells are found in cancer-survivors subjected to chemotherapy and radiotherapy in the course of their treatments[5-8]. To verify the impact of senescence in the context of cancer immunotherapy, we used two validated mouse models of senescence induction. One was generated by total body irradiation (TBI) at the sub-lethal dose of 6.5 Gy and the second was induced by a single injection of doxorubicin at a dose of 10 mg/kg (Doxo; Fig. 1a). These models are relevant as cancer patients are often aged and/or have been previously exposed to cytotoxic drugs prior to their immunotherapy treatments. We and others have shown that the senescence phenotype is established gradually in these mouse models and remains stable at least 8-10 weeks after TBI[5] and 4 weeks after the injection of Doxo[11]. Indeed, the expression of *p21* and/or *p16* is increased in the spleen of TBI or Doxo-treated mice (Fig. 1b). Importantly, hematopoietic cell counts in blood can fully recover in both models (Supplementary Fig. 1). Moreover, to investigate the specific role of senescent cells we took advantage of the p16-3MR mouse model which allows the tracking and the elimination of p16+ senescent cells following the injection of ganciclovir (GCV)[25]. As expected from our previous work, exposure of p16-3MR mice treated with GCV leads to the elimination of p16+ cells in the spleen (Fig. 1c)[12]. Hence, to evaluate the impact of senescence on the efficacy of αPD-L1 immunotherapy treatments we injected MC38 tumor cells in p16-3MR mice previously exposed to TBI or Doxo. The injection of the αPD-L1 antibody limited the growth of MC38 tumors and improved the survival of control (CTL) mice. However, the same treatments in both TBI and Doxo treated mice failed restraining tumor growth and did not improve survival (Fig. 1d–f). These observations were not limited to the MC38 cell line as the efficacy of αPD-L1 treatments was also diminished in TBI-treated mice injected with the EL4 lymphoma cell line (Supplementary Fig. 2)[26,27]. Importantly, the elimination of p16+ cells by the injection of GCV could restore the efficacy of αPD-L1 and improve survival in mice previously exposed to TBI (Fig. 1d–f). Note that as expected, GCV had no beneficial effect in Doxo-treated mice as p16 expression was not increased in these animals, a phenotype also observed by others[28]. Intriguingly, tumor growth was delayed in mice previously exposed to TBI or Doxo compared to control mice, a phenotype also observed in aged mice[29,30]. Tumor growth rate was not changed by treatments with GCV alone as TBI-treated mice still died around 40 days after the injection of tumor cells compared to approximately 25 days in the case of control mice, suggesting tumor growth delay is independent of senescence. Altogether, these results demonstrate the detrimental impact of senescent cells on the tumor immune response and that their removal can restores the efficacy of immunotherapy.

### Locally induced senescence is sufficient to decrease the efficacy of immunotherapy

We next asked if senescence induced locally in the context of TIS would also compromise the efficacy of αPD-L1 treatments. To answer this question, we used a model where established MC38 tumors (approximately 50mm³ in size) were exposed to a single 12 Gy dose of local radiotherapy (RT) (Supplementary Fig. 3A). It is possible to only expose the tumor to RT by shielding the remaining of the mouse with a led plate. At this dose, senescence was induced in tumors as determined by expression of SA-βGal and p16-dependent bioluminescent signal associated with the Renilla-luciferase expression of senescent cells in p16-3MR mice (Supplementary Fig. 3B, C). Senescence induction coincided with the tumor growth being temporarily halted for more or less 14 days after which growth resumes. Hence, 14 and 17 days after local RT, mice were injected with an αPD-L1 antibody and tumor growth monitored. As we observed in mice previously exposed to TBI and Doxo, locally induced senescence is sufficient to decrease the efficacy of αPD-L1 treatments (Supplementary Fig. 3D).

### Elimination of senescent cells using ABT263 restores immunotherapy efficacy in mice

In an effort to provide a more translational approach, we next wanted to remove senescent cells using a pharmacological treatment and see if it would also improve the success of immunotherapy. As such, we evaluated if ABT263, a validated senolytic drug[18], would efficiently clear senescent cells in TBI-treated mice. For that purpose, we monitored the bioluminescent signal associated with senescent cells in p16-3MR mice. We observed a decreased luciferase signal and p16/p21 gene expression levels, from the spleen of p16-3MR TBI mice treated with ABT263 (Fig. 2a–c). As observed with GCV, ABT263 treatment improved the efficacy of αPD-L1 immunotherapy against MC38 tumor cells in TBI mice (Fig. 2d, e). In addition, ABT263 significantly prolonged the survival of mice previously exposed to TBI while it had no impact on control mice (Fig. 2f). To understand how senescent cell clearance improved anti-tumor immune response, we used flow cytometry to evaluate immune cell infiltration in tumors dissociated 2 days following the last αPD-L1 injection (day 14 post tumor inoculation). Interestingly, we observed a higher proportion of CD8 T cells in tumors of all groups receiving αPD-L1 injections except in tumors dissociated from TBI mice (Fig. 2g). There was no significant differences in CD4 T cells and a downward trend in the proportion of myeloid cells (CD11b+) in all groups receiving αPD-L1 injections except in TBI mice not treated with ABT263 (Fig. 2g). This is in accordance with a previous report showing that CD8 T cells significantly proliferate in response to ICIs therapy[31]. Consequently, the ratio of CD8 T to CD11b + cells was increased after immunotherapy in all groups excluding the group of TBI mice not treated with ABT263 (Fig. 2h). Importantly, ABT263 had no significant impact on the tumor growth rate and on the immune infiltrate in absence of αPD-L1 treatments. These observations suggest that senescence, either directly or through its impact on the TME, interferes with the efficacy of αPD-L1 immunotherapy by limiting the expansion of CD8 T cells within tumors.

### ABT263 improves the abscopal effect

We next wanted to evaluate if senescence would also compromise the immune response involved in the abscopal effect. The abscopal effect is defined by the immune clearance of a secondary tumor distant from a primary tumor treated by RT. When associated with ICIs, RT was shown to greatly improve the abscopal effect presumably by helping with the release and presentation of tumor antigens[32]. We investigated the abscopal effect by inoculating mice on each flank with MC38 tumor cells, a primary tumor and a smaller secondary tumor. The protocol we used involved exposing the primary tumor with a fractionated dose of RT (3x8Gy) followed by CTLA-4 injections on days 14, 17, and 20 (Fig. 3a). Here again, it is possible to only expose the primary tumor to

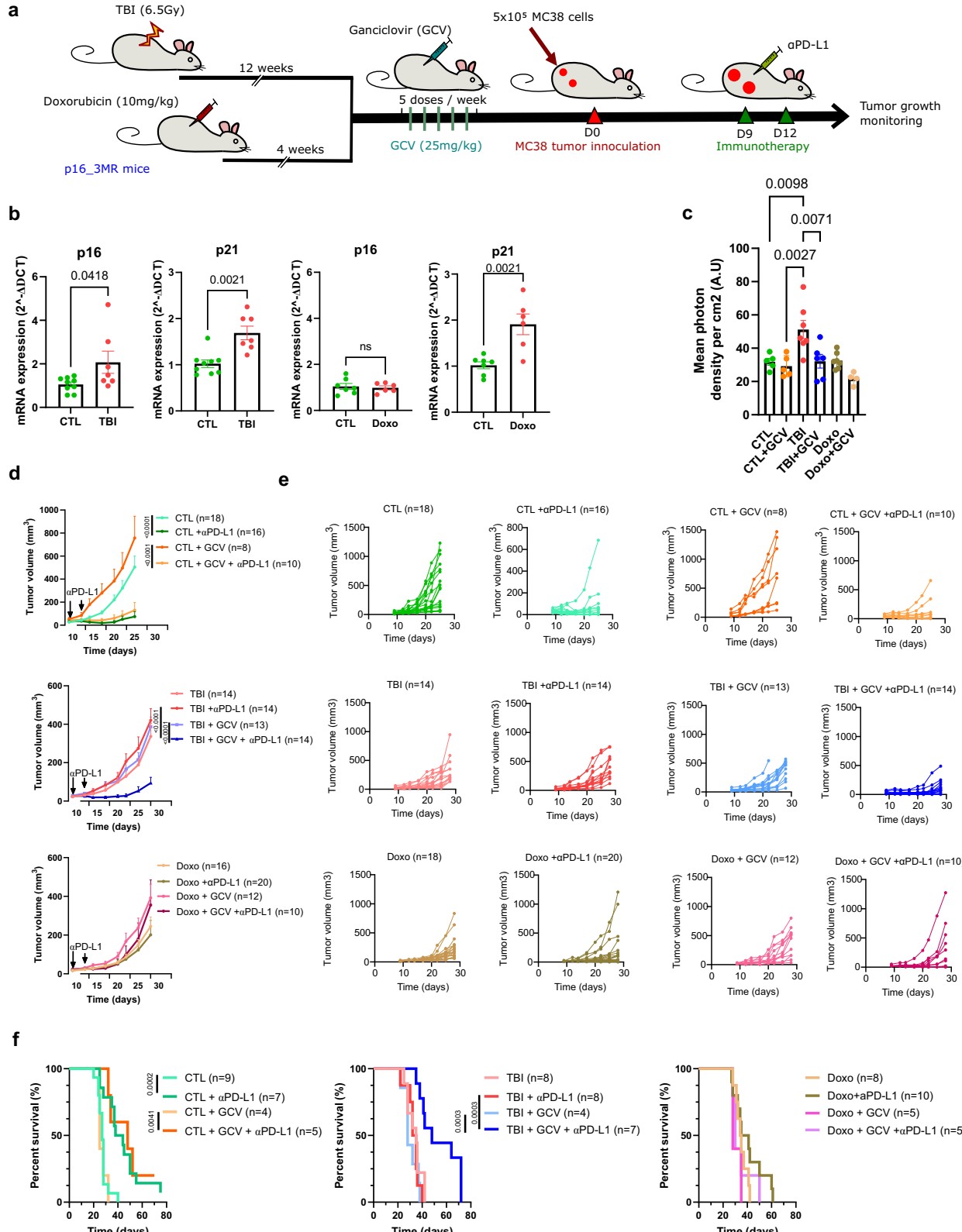

RT by shielding the remaining of the mouse with a led plate. For this experiment, we choose to use a blocking αCTLA-4 antibody since its injection alone was shown to have no effect on the growth of MC38 tumors while its combination with RT greatly improves the abscopal effect[33]. Note that in this model, the primary tumor is almost completely eliminated in response to RT alone in both control and TBI mice. However, the immune clearance of the secondary abscopal

tumor is entirely dependent on αCTLA-4 treatment. Using this combination, we observed that the abscopal effect allowed for almost the complete elimination (CE) of the secondary tumor in control mice while the efficacy was reduced in TBI mice (Fig. 3b, d, e). The injection of mice with ABT263 prior to RT enhanced the efficacy of the abscopal effect in TBI mice and improved their survival to a level similar to what we observed in control mice (Fig. 3b, c). Here again, we investigated

**Fig. 1 | Senescence leads to immunotherapy resistance. a** Schematic of the experiment. In brief, senescence was induced in p16-3MR mice with a single 6.5 Gy sub-lethal dose of total body irradiation (TBI) or following the injection of Doxo (10 mg/kg). 12 weeks after TBI or 4 weeks after doxorubicin injection, mice were injected intraperitoneally with GCV (25 mg/kg) for 5 consecutive days to remove senescent cells. 5 days after the last injection of GCV, mice were injected on each flank with $5 \times 10^5$ MC38 tumor cells expressing mPlum fluorescent protein. Mice were injected with a αPD-L1 blocking antibody on days 9 and 12 and tumor growth was evaluated until reaching a limit point. **b** Relative expression of *p16* and *p21* was quantified by RT_qPCR in the spleen 10 weeks after induction of senescence with TBI or 4 weeks after Doxo treatments. *n* = 9 CTL and 6 TBI; *n* = 7 and *n* = 6 Doxo from independent animals. Two-tailed unpaired *T*-test for *p16/p21* in Doxo group, and Mann-Whitney test for *p16/p21* in TBI group. Shown is the mean ± SEM. **c** p16 expression measured by bioluminescence in the spleen of p16-3MR mice before and after treatment with GCV. *n* = 5 CTL, *n* = 5 CTL + GCV, *n* = 7 TBI, *n* = 6 TBI + GCV, *n* = 7 Doxo, *n* = 4 Doxo+GCV from independent animals. Ordinary one-way ANOVA with Tukey correction. Shown is the mean ± SEM. **d** Shown is the mean tumor growth±SEM over 25 days for CTL mice and 28 days for TBI/Doxo mice. Control mice that were not subjected to TBI or Doxo are indicated as CTL. The total number of tumors per group is indicated in parenthesis. Mixed-effect analysis or Two-way ANOVA with Tukey correction. **e** Shown is the size of each individual tumors at the indicated timepoint. The total number of tumors per group is indicated in parenthesis. **f** Kaplan-Meyer survival plots of CTL, TBI and Doxo mice for each of the indicated conditions. The total number mice per group is indicated in parenthesis. Log-rank (Mantel-Cox test). Source data are provided as a Source Data file.

the immune infiltration in the TME by dissociating secondary tumors on day 22 post tumor inoculation or two days after the last injection of αCTLA-4. At this timepoint, the abscopal effect is at his strongest and secondary tumors start to be rejected in all groups. Flow cytometry analysis revealed that the proportion of CD8 T cells infiltrating the TME after RT + αCTLA-4 treatments is higher in control compared to TBI mice (Fig. 3f). The proportion of myeloid CD11b+ cells decreased following RT + αCTLA-4 treatments in all groups although more modestly in TBI mice (Fig. 3f). Consequently, the ratio of CD8 T to CD11b+ was lower in tumors dissociated from TBI mice that did not receive ABT263 (Fig. 3g). Inversely, the injection of ABT263 increased the proportion of CD8 T cells in TBI mice treated with RT+aCTLA-4 and improved the ratio to myeloid cells. ABT263 had no significant impact on the tumor growth rate and on the immune infiltrate in absence of immunotherapy treatments. Finally, we observed important differences in the proportion of CD8 T cells expressing IFNγ in tumors of control vs TBI mice treated with RT + αCTLA-4. Again, treatment of TBI mice with ABT263 restored the proportion of CD8 T cells expressing IFNγ (Fig. 3h). These results reinforce the notion that the immune response to ICIs therapy is compromised by senescence and that the effect is reversible upon treatment with a senolytic drug.

## Monocytes with an immunosuppressive phenotype accumulate in the TME of TBI mice

Many distinct mechanisms may be responsible for the reduced accumulation of CD8 T cells in the TME of TBI mice. For one, increased expression of immunosuppressive ligands at the surface of tumor cells may be implicated[34]. However, no difference in the expression of Galectin9, PD-L1, Tnfrs4, and CD86 expression was detected by flow cytometry at the surface of dissociated tumor cells at day 22 post inoculation in our different groups (Supplementary Fig. 4). Another explanation may reside in the capacity of splenocytes to process and present antigens to T cells. As such, we measured the capacity of antigen-presenting cells (dendritic cells and macrophages here after referred as APC) collected from the spleen of control and TBI mice. We observed no difference in the capacity of these cells to phagocyte and process the OVA-DQ protein (Supplementary Fig. 5A). Similarly, the capacity of APC collected from TBI mice to cross-present the GP-33 peptide to CD8 T cells expressing a GP-33-specific TCR was not diminished (Supplementary Fig. 5B). However, we cannot rule out that APC in lymph nodes or within tumors were not impaired. Then, we also asked if the migration and infiltration potential of CD8 T cells collected from TBI mice was compromised. We found these cells were not incapacitated in their ability to infiltrate MC38 spheroid in vitro (Supplementary Fig. 5C). Combined with our previous observation that purified CD8 T cells from TBI mice can proliferate normally upon stimulation[12], these results suggest that an immunosuppressive TME is likely responsible for the reduced accumulation of CD8 T cells within the tumor.

We thus performed an unbiased analysis of the TME composition using single-cell transcriptomic analysis of cells dissociated from tumors collected from control (Ctrl), TBI and TBI + ABT263 (ABT)

treated mice. An average of 10,000 cells collected and pooled from 3 different tumors in each group were analyzed (Fig. 4a). All cells were clustered and embedded in two dimensions using UMAP representation (Fig. 4b). All clusters were identified and named based on a list of classically used markers to identify immune cells (Fig. 4c and Supplementary Data 1). First observation is that the size of the cluster representing monocytes (Itgam+, Ccr2High, Cx3cr1low, Ly6c2High, Adgre1low) tended to increase in tumors of TBI mice. On the other hand, the size of the clusters representing CD8 T cells (Cd3g+, Cd8a+), natural killer (NK, Itgam-, Nkg7+, Klrb1c+), dendritic cells (DC, Itgax+) and DC expressing Xcr1 (DC Xcr1, Itgax+ Xcr1+) tended to be smaller in the TBI group. Finally, the clusters representing CD4 T cells (Cd3g+, Foxp3+, Cd4+), B cells (Cd3-, Itgam-, Ly6d+) and macrophages (Itgam+, Ly6c2low, Ccr2low, Cxc3cr1high, Adgre1high) remained mostly unchanged (Fig. 4d).

We next analyzed the differently expressed genes (DEGs) within major cell clusters in the TBI group compared to the same cell population in the control or ABT groups (Supplementary Data 2). By using UpSetPlot representations, to identify intersection (shared DEGs) between groups, we found that monocytes and macrophages exhibit the highest number of DEGs in TBI vs Ctrl (290 and 228 respectively) and in ABT vs TBI (123 and 171 respectively), meaning that these cell types are those most affected by TBI (Fig. 4e). More importantly, 63 DEGs (21% of 290) in monocytes and 26 DEGs (11% of 228) in macrophages are in the intersection of Ctrl vs TBI and ABT vs TBI groups, indicating the expression of those genes are fully restored after ABT263 treatments (Fig. 4e and Supplementary Fig. 6). We first focused on monocytes since those cells contained the highest number of restored DEGs and represent the largest cluster. Of the 63 DEGs in monocytes, we generated a signature of 12 overexpressed genes exhibiting immunosuppressive functions and/or associated with a poor cancer prognosis: Cd44[35], Cd83[36], Emp1[37], Eif4e[38], Thbs1[39], Vegfa[40], Emilin2[41], Tgm2[42], Ptgs2[43], Fn1[44], Crem[45], Il1rn[46] (Fig. 4f). We referred that combination of gene as the TBI-Monocyte-Signature, and its global expression was quantified (Fig. 4f lower panel). In addition we observed the downregulation of Atf3, a transcription factor whose the downregulation is associated with a poor tumor outcome[47] (Fig. 4f). FeaturePlot representations confirmed this gene signature is indeed associated with monocytes (Fig. 4g). We also found a similar gene signature in macrophages, especially for the downregulation of Atf3 and the upregulation of Crem, Cd83, Cd44, Tgm2 and Thbs1 which were restored by ABT263 (Supplementary Fig. 7). Finally, we then compared the TBI-Monocyte-Signature with a previously published transcriptomic data set generated from B16F10 tumors extracted from young (3 months) and old mice (18–20 months)[30]. Using the metadata generated by the authors for monocytes (Ly6c2 + , Ccr2 +) we found that the TBI-Monocyte-Signature is overexpressed in old monocytes vs young monocytes, suggesting that exposure to TBI induces a premature aging phenotype (Fig. 4h). Among the 63 DEGs founds in monocytes, none of them are associated with the classic patterns of senescence process such as DNA damage, CDKi expression or the senescence-associated secretory phenotype. This observation suggests that monocytes are not senescent in the tumor microenvironment.

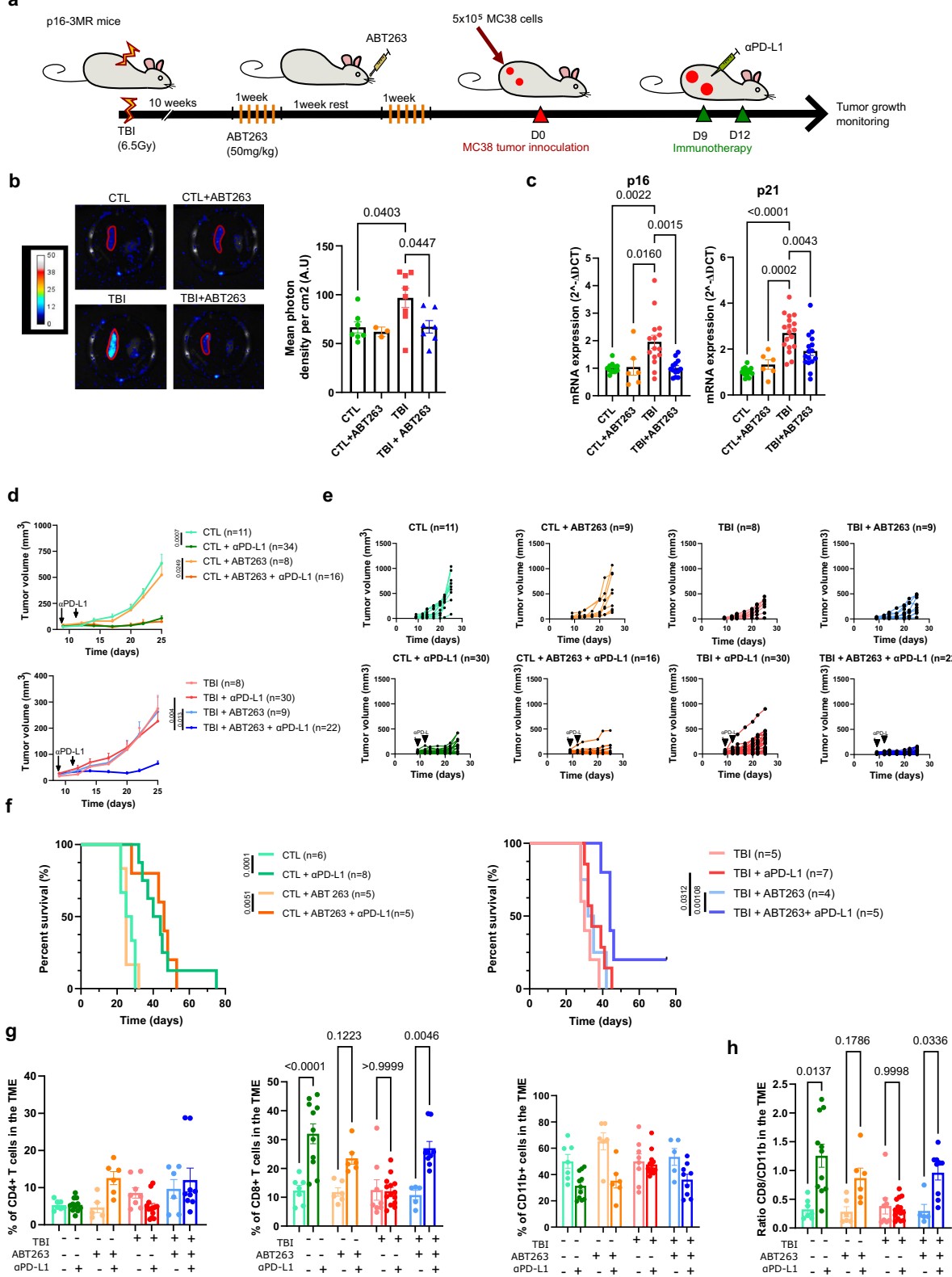

## Depletion of myeloid cells partially restores the efficacy of αPD-L1 treatments

Our results so far suggest that the immunosuppressive profile of monocytes and macrophages are responsible for the impaired accumulation of CD8 T cells in tumors of TBI mice. Indeed, GSEA performed on DEGs from CD8 T cells showed a reduced activation state of CD8 T cells collected from TBI mice compared to Ctrl (Fig. 5a, b). GSEA also

showed restored activation of CD8 T cells after ABT263 treatments (Fig. 5a, b). In line with these observations, the expression of the effector function genes Ifng and Cd69 tended to decreased in CD8 T cells collected from TBI tumors. (Fig. 5c). Therefore, we designed an in vitro assay to measure the impact of myeloid CD11b+ cells on the proliferation of CD8 T cells. As such, CD8 T cells were collected from dissociated tumors and activated with IL2 and CD3/CD28 beads in

**Fig. 2 | Senescent cells elimination using ABT263 treatments improves αPD-L1 cancer immunotherapy in irradiated mice. a** Schematic of the experiment. In brief, senescence was induced using sub-lethal TBI and 10 weeks later mice received ABT263 (50 mg/kg), by intragastric gavage for two cycles of 5 consecutive days, to remove senescent cells. 5 days after the last gavage, mice were either sacrificed to measure senescence in spleen ex-vivo or injected with MC38 tumor cells. **b** Evaluation by bioluminescence of p16 expression in the spleen before and after treatment with ABT263. Shown is the mean ± SEM. $n = 7$ CTL, $n = 3$ CTL + ABT263, $n = 8$ TBI, $n = 7$ TBI + ABT263 from independent animals. Ordinary one-way ANOVA with Tukey correction. **c** Evaluation by qPCR of p16 and p21 expression in the spleen before and after treatment with ABT263. $n = 11$ CTL, $n = 6$ CTL + ABT263, $n = 15$ TBI, $n = 13$ TBI + ABT263 from independent animals. Ordinary one-way ANOVA with Tukey correction. Shown is the mean ± SEM. **d** Tumor growth was evaluated for each of the indicated groups of mice. Each line represents the mean growth±SEM of tumor over 25 days or until the mouse had to be removed from the study. The total number of tumors per group is indicated in parenthesis. Mixed-effect analysis or Two-way ANOVA with Tukey correction. **e** Shown is the size of each individual tumors at the indicated timepoint. Mice that were not subjected to TBI are indicated as control (CTL). The total number of tumors per group is indicated in parenthesis. **f** Kaplan–Meyer survival plots of both control and TBI mice for each of the indicated conditions. Log-rank (Mantel-Cox test). **g** Tumor immune cells infiltration was analyzed by flow cytometry after tumor dissociation on day 14 post inoculation. Shown is the proportions of CD4 T cells, CD8 T cells and CD11b+ cells within the CD45+ population. $n = 7$ CTL, $n = 11$ CTL (αPD-L1), $n = 6$ CTL + ABT263 (αPD-L1), $n = 6$ CTL + ABT263 (αPD-L1), $n = 8$ TBI, $n = 14$ TBI (αPD-L1), $n = 5$ TBI + ABT263, $n = 9$ TBI + ABT263 (αPD-L1) tumors per group. Ordinary one-way ANOVA with Tukey correction. Shown is the mean ± SEM. **h** Shown is the ratio of the CD8 T cells over CD11b+ in tumors. $n = 7$ CTL, $n = 11$ CTL (αPD-L1), $n = 6$ CTL + ABT263 (αPD-L1), $n = 6$ CTL + ABT263 (αPD-L1), $n = 8$ TBI, $n = 14$ TBI (αPD-L1), $n = 5$ TBI + ABT263, $n = 9$ TBI + ABT263 (αPD-L1) tumors per group. Brown-Forsythe and Welch ANOVA with Games-Howell correction. Shown is the mean ± SEM. Source data are provided as a Source Data file.

presence or after removal of CD11b+ cells. We observed that CD8 T cells collected from tumors of TBI mice proliferated less upon stimulation suggesting that CD11b+ cells were responsible for this inhibition (Fig. 5d). Finally, we depleted Ly6c+ monocytes in TBI mice using a blocking antibody and measured if this would bypass resistance to αPD-L1 treatments (Fig. 6a). We first confirmed the depletion of Ly6c + /CD11b+ positive cells in blood of TBI mice (Fig. 6b). Then, we observed that only the combination of Ly6c and αPD-L1 blocking antibodies could efficiently delay tumor growth in TBI mice (Fig. 6c, d). These results demonstrate that myeloid cells in the context of a senescent environment have an exacerbated immunosuppressive profile and induce resistance to immunotherapy.

## Discussion

This work revealed that the accumulation of senescent cells interferes with cancer immunotherapies and that their removal, using a genetic or pharmacological approach, reverses the phenotype. Mechanistically, we showed that senescence, through its action on myeloid cells, compromises the efficacy of immunotherapy by limiting the accumulation of CD8 T cells within the TME. These results in combination with our previously published data showing that T cells purified from TBI mice can be activated and expanded at a level similar to cells collected from control mice[12], suggest that the effect on T cells is mostly mediated in a non-cell autonomous manner. Moreover, we demonstrated that T cells from TBI mice are efficiently primed by antigen-presenting cells (dendritic cells and macrophages) and are still capable of infiltrating MC38 tumor cell spheroids in vitro. Instead, it appears that the deleterious effect arises from the lower ratio of CD8 T cells/myeloid cells (Cd11b + ) observed in TBI mice after immunotherapy. A recent publication showed that αPD-L1 treatments lead to increase apoptosis of myeloid cells[48]. We did not observe more myeloid cells death in TBI mice treated with αPD-L1. Yet, because the proportion of CD8 T cells was decreased in the TME of TBI mice, this suggested that myeloid cells, despite not significantly more abundant, were functionally more immunosuppressive. We confirmed this phenotype using both single-cell transcriptomic data and functional assays in vitro and in vivo.

Others have shown that infiltrating monocytes expressing high levels of Thbs1 and Vegfa, two genes from our TBI-induced Immunosuppressive signature, can inhibit the efficacy of αPD-L1 treatments against Lewis lung carcinoma tumors[49]. Single-cell transcriptomic analysis of both the spleen and BM of control and TBI mice suggest that monocytes were already partially polarized by the systemic senescent environment in absence of tumor cells (Supplementary Figs. 8 and 9). However, this is not a prerequisite as we also showed that local RT in the context of TIS is sufficient to impair the efficacy of αPD-L1 treatments (Supplementary Fig. 3). Of the many immunosuppressive genes we observed, Thbs1 is a secreted glycoprotein implicated in a wide range of processes[50]. Blocking Thbs1 with a genetic approach in triple-negative breast cancer (TNBC) was shown to increase CD8 T cells infiltration/proliferation and improve response to αPD-L1 immunotherapy in mice[51]. To our knowledge, there is no validated pharmacological inhibitor of Thbs1 available. Similarly, increase Vegfa expression by monocytes is likely detrimental since it was shown that antiangiogenics molecules improve the efficacy of immunotherapies[52]. Another gene from our signature is Atf3, whose expression is downregulated in monocytes collected from TBI tumors. Atf3 is a gene known to play a crucial role in several biological processes, such as immune response regulation. It has been reported that its downregulation in macrophages is associated with a worse outcome in patients with hepatocellular carcinoma[47]. Atf3 was also shown to be downregulated in hematopoietic stem and progenitor cells during aging, and its expression restored after a parabiosis experiment with young mice[53]. It remains to be determined if a correlation can be drawn between immunotherapy efficacy and Atf3 expression after TIS. Finally, a recent study showed that IL-6 expression can help predict the response to αPD-L1 therapy[54]. However, in our study, IL-6 expression was not changed between groups. Overall, we believe the immunosuppressive signature and its impact on myeloid cells is likely multifactorial.

Noteworthy, in our abscopal model we observed that the elimination of the primary tumor was not compromised by senescence. This suggest that fractionated doses of RT were sufficient to kill cancer cells, likely through DNA damage-induced apoptosis, and that RT was not inhibited by the immunosppressive TME. Likewise, given the growing literature linking the accumulation of senescent cells with age-associated diseases, it is tempting to speculate that senolytics may also be able to increase ICB efficacy in aged subjects. Indeed, decrease immune functions in aged mice was reported to be associated with a decline in ICB efficacy[55–57]. This is in contrast to studies in humans who did not show a correlation between decrease ICB efficacy and aging[58]. For example, no decrease in the efficacy of ICB was observed in aged (over 70 years old) lung cancer patients[59]. However, in this study all patients (younger and old) were previously treated with platinum chemotherapy prior to receiving ICB therapy. Given platinum was shown to induce senescence it is possible that both groups had a compromised immune response independently of their age at the time of ICB treatments. For this study, we choose to use ABT263 because of its proven efficacy at eliminating senescent cells in mice. However, ABT263 is known to induce thrombocytopenia in patients and therefore has limited clinical utility[60]. For our approach to be eventually tested in patients, one would likely need to employ a different senolytic approach. It may be possible to use other proven senolytics such as quercetin+dasatinib[61] and fisetin[62], assuming they can efficiently improve immune cell functions.

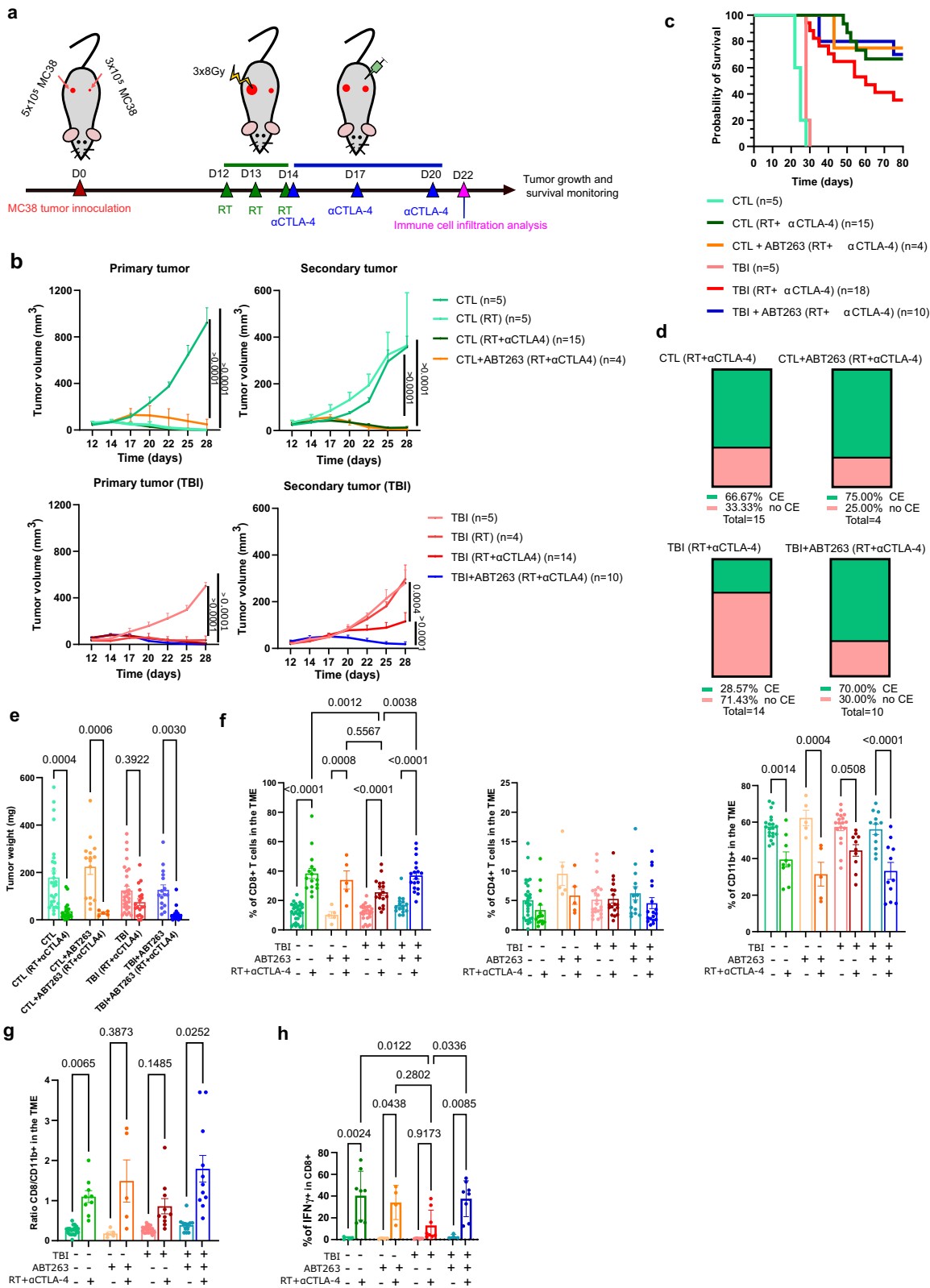

Overall, our results demonstrate that it is possible to pharmacologically increase the tumor immune response and suggest senolytic drugs injected prior to ICB should be considered. We also identified an immunosuppressive signature in myeloid cells that it may be possible to exploit to predict the efficacy of immunotherapy. We foresee that cancer treatments that do not induce DNA damage and cellular senescence will not interfere with ICB efficacy.

## Methods

### Animals

p16-3MR mice on a C57BL/6 background were kindly donated by Dr. Judith Campisi (Buck Institute) according to a material transfer agreement. All mice were bred on-site and in vivo manipulations were approved by the Comité Institutionnel des Bonnes Pratiques Animales en Recherche of the CHU Ste-Justine (protocol 2023–5331). Both males

**Fig. 3 | Impaired abscopal effect in previously irradiated mice is restored by ABT263. a** Schematic of the experiment. CTL or TBI mice treated or not with ABT263 were inoculated on each flank with 3 or $5 \times 10^5$ MC38 tumor cells. On days 12, 13, and 14 after tumor inoculation, the larger primary tumor was exposed to 3 rounds (8 Gy each) of radiotherapy (RT). Mice were then injected on days 14, 17, and 20 with a αCTLA-4 blocking antibody and tumor growth evaluated until reaching a limit point. Alternatively, some mice were euthanized at day 22 and tumors surgically removed to allow immune infiltration analysis. **b** Tumor growth was evaluated for each of the indicated groups of mice. Each line represents the growth of an individual tumor over 28 days or until the mouse had to be removed from the study. The total number of tumors per group is indicated in parenthesis. 2way ANOVA with Tukey correction. Shown is the mean ± SEM. **c** Kaplan–Meyer survival plots for each of the indicated conditions. **d** Histogram showing the proportion of mice in each group achieving complete secondary (abscopal) tumor elimination (CE). **e** Shown is the weight of the secondary tumors at day 22 prior to dissociation. $n = 29$ CTL, $n = 17$ CTL (RT + αCTLA4), $n = 15$ CTL + ABT263, $n = 5$ CTL + ABT263 (RT + αCTLA4), $n = 29$ TBI, $n = 19$ TBI (RT + αCTLA4), $n = 16$ TBI + ABT263, $n = 19$ TBI + ABT263 (RT + αCTLA4) tumor from independent animals. Brown-Forsythe and Welch ANOVA test with Games-Howell correction. Shown is the mean ± SEM. **f** Tumor immune cells infiltration as determined by flow cytometry from

dissociated tumors on day 22 post inoculation. Shown are the proportions of CD8 T cells, CD4 T cells and CD11b+ cells over the total number of CD45+ cells collected from tumors. For CD8 and CD4, $n = 29$ CTL, $n = 17$ CTL (RT + αCTLA4), $n = 5$ CTL + ABT263, $n = 5$ CTL + ABT263 (RT + αCTLA4), $n = 24$ TBI, $n = 17$ TBI (RT + αCTLA4), $n = 16$ TBI + ABT263, $n = 18$ TBI + ABT263 (RT + αCTLA4) tumors from independent animals. For CD11b, $n = 20$ CTL, $n = 9$ CTL (RT + αCTLA4), $n = 5$ CTL + ABT263, $n = 5$ CTL + ABT263 (RT + αCTLA4), $n = 18$ TBI, $n = 10$ TBI (RT + αCTLA4), $n = 12$ TBI + ABT263, $n = 11$ TBI + ABT263 (RT + αCTLA4) tumors from independent animals. Ordinary one-way ANOVA with Tukey correction. Shown is the mean ± SEM. **g** Shown is the ratio of CD8 T cells over CD11b+ cells in tumors at day 22. $n = 20$ CTL, $n = 9$ CTL (RT + αCTLA4), $n = 5$ CTL + ABT263, $n = 5$ CTL + ABT263 (RT + αCTLA4), $n = 18$ TBI, $n = 10$ TBI (RT + αCTLA4), $n = 12$ TBI + ABT263, $n = 11$ TBI + ABT263 (RT + αCTLA4) tumors from independent animals. Brown-Forsythe and Welch ANOVA with Games-Howell correction. Shown is the mean ± SEM. **h** Shown is the quantification of the expression of IFNγ as measured by flow cytometry in infiltrated CD8 T cells at day 22. $n = 4$ CTL, $n = 8$ CTL (RT + αCTLA4), $n = 4$ CTL + ABT263, $n = 4$ CTL + ABT263 (RT + αCTLA4), $n = 4$ TBI, $n = 8$ TBI (RT + αCTLA4), $n = 4$ TBI + ABT263, $n = 8$ TBI + ABT263 (RT + αCTLA4) tumors from independent animals. Ordinary one-way ANOVA with Tukey correction. Shown is the mean ± SEM. Source data are provided as a Source Data file.

and females mice were used as we did not observed difference in tumor growth and response to immunotherapy between sex.

### Senescence mouse models
12- to 14-week-old mice were exposed to TBI at the single sub-lethal dose of 6.5 Gy (1 Gy/min) using a Faxitron CP-160. During the 10 days following TBI, Baytril® antibiotic was added to the water to prevent infections. Alternatively, doxorubicin (CHU Sainte-Justine-Pharmacy) was administrated with a single intraperitoneal injection at a dose of 10 mg/kg.

### In vivo treatments
10 to 12 weeks after TBI or 1 week after doxorubicin, mice received intragastric gavage with vehicle alone (ethanol: polyethylene glycol 400:Phosal 50 PG at 10:30:60) or with ABT263 (MedChemExpress, Navitoclax) at 50 mg/kg/body weight. Vehicle and ABT263 was given to mice for two cycles of 5 consecutive days, with a week of rest between cycles. GCV (CHU Sainte-Justine-Pharmacy), was injected 10–12 weeks after TBI. GCV was administrated intraperitoneal for 5 consecutive days at a dose of 25 mg/kg in PBS (Multicell) after activation in 0.1 M of hydrochloric acid (Sigma).

### Cell culture
MC38 and EL-4 cells were grown in DMEM or RPMI1640 respectively (Multicell) supplemented with 10% FBS and 1% penicillin-streptomycin at 37 °C under 5% CO2 in a humidified atmosphere. MC38 tumor cells were transduced with a lentivector to express mPlum[63] and sorted to obtain a nearly pure population with high mPlum fluorescence. Cells are routinely tested to ensure the absence of mycoplasma contamination.

### Tumor inoculation and immunotherapy treatments
MC38 and EL-4 tumor cells (were a gift from Dr. John Stagg and Dr. Hélène Decaluwe respectively) were inoculated subcutaneously ($5 \times 10^5$ and $3 \times 10^5$ cells in PBS respectively) in both the right and left flanks of mice anesthetized with isoflurane. A blocking αPD-L1 antibody (BioXcell, clone 10 F.9G2) was injected intraperitoneally (200 μg in PBS per mouse) on day 9 and 12 post tumor cells inoculation when tumor size was around 25–30mm³. Blocking Ly6c antibody (BioXcell clone Monts 1) was injected intraperitoneally (200 μg in PBS per mouse) at day 7, 11 and 15. For the abscopal assay, MC38 cells were inoculated with two different concentrations, $5 \times 10^5$ cells at the primary tumor site (right flank) and $3 \times 10^5$ cells in the secondary tumor site (left flank). The primary tumor was treated with three doses of radiotherapy ($3 \times 8$ Gy)

on days 12, 13, and 14. A blocking αCTLA-4 antibody (BioXcell, clone 9H10) was injected intraperitoneally (200 μg in PBS per mouse) on day 14, 17, and 20 post tumor cells inoculation when the size of the primary tumor was between 45–50 mm³ and the secondary tumor between 20–25mm³. Tumor growth was monitored by fluorescence acquisition (Labeo Technologies) and caliper measurement. For survival analysis, mice were sacrificed when one limit point was reached according to our animal comity guidelines. Our comity established limit points as no more than 10% weight loss, no destress signs such as alopecia or decreasing activity, MC38 tumor size does not reach more than 1cm³ without ulceration or 0.5cm³ with ulceration, EL-4 tumor size does not reach more than 2cm³. In some cases, one of these limit point has been exceeded the last day of measurement and the mice were immediately euthanized. Tumors, spleen, and BM were excised after mice were sacrificed by lethal intraperitoneal injection of a barbituric followed by neck dislocation.

### Tissues dissociation
Tumors were collected and dissociated with the mouse tumor dissociation kit (Tumor dissociation kit for mice, Miltenyi). We used the program 37C_m_TDK2 in an Octo-Dissociator with heaters (Miltenyi). After dissociation, the enzymatic reaction was stopped by adding 10 mL of RPMI1640 supplemented with 10% FBS and the cell suspension filtered through a 70 μm mesh. The cells were then ready for analysis or frozen in nitrogen liquid for later usage. Splenocytes were obtained by crushing the spleen between a syringe cap and then filtered with a 70 μm mesh. Red blood cells were removed by adding for 2 min a Red Blood cell lysis solution (NH4Cl 0.8 mg/L, NaHCO3 0.84 mg/L, and EDTA 0.37 mg/L). Cells from the BM were obtained after flushing with a needle the soft center of tibias and femurs.

### Flow cytometry
Cells were blocked with PBS containing 2% FBS and stained using antibodies against CD16/CD32 (1/100 dilution from Biolegend) for 30 min at 4 °C. Cells were then washed in PBS containing 2% FBS and stained with conjugated antibodies for cytometry during 20 min at 4 °C in the dark (1/100 dilution). Cells were washed again and dead cells stained with the Zombie staining dye (Biolegend) for 15 min at room temperature in PBS without FBS. Cells were washed and processed on a Fortessa (BD Biosciences) for analysis. For the analysis of the TME, dissociated cells were stained using the following antibodies: CD45 (CD45-BV785, Biolegend), CD3 (CD3-AF700, Biolegend), CD4 (CD4-PE, Biolegend), CD8 (CD8-PercpCy5, Biolegend), CD11b (CD11b-BUV395, Biolegend), tumor cells were self-fluorescent (mPlum). Dead cells were

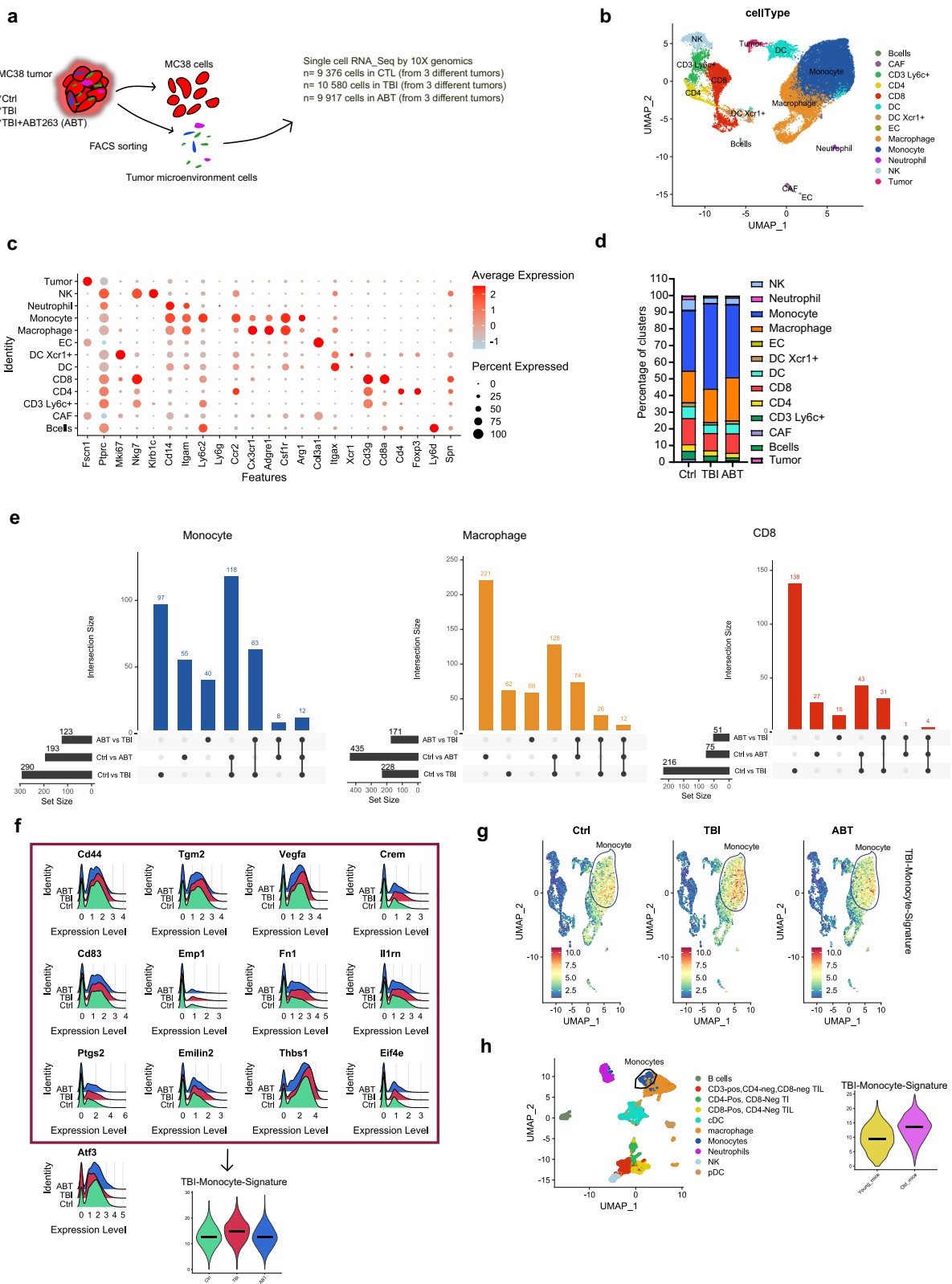

stained with the Zombie-Aqua dye (Biolegend) and then analyzed by flow cytometry. For the staining of IFNγ, cells were treated with Momensin-A 1X (BioLegend) and with PMA-Ionomicine (Invitrogen™ eBioscience™ Cell Stimulation Cocktail 500X) for 2 h at 37 °C under 5% $CO_2$ in a humidified atmosphere in complete RPMI1640. Cells were then fixed and permeabilized with the BD Cytofix/Cytoperm™ Kit and stained with an anti IFNγ antibody (IFNγ − BV785, Biolegend). The full

list of antibodies used in this study is available in Supplementary Data 3. All gating strategy are presented in Supplementary Fig. 10.

### SA-βGal staining
Whole tumors were isolated and fixed in 4% PFA for 20 min. After two washes in PBS, the whole tumor was soaked in β-Gal staining solution at 37 °C (1 mg/mL X-Gal, 40 mM Citric acid Sodium phosphate buffer at

**Fig. 4 | The immunosuppressive phenotype of monocytes is partially restored after treatment with ABT263. a** Tumors from the indicated groups of mice were collected at day 22 post inoculation and dissociated. Non tumorigenic cells (referred as tumor microenvironment cells) were sorted and used for single-cell transcriptomic analysis in two independent experiments. Each group contains a pool of cells from three tumors. Approximately 10,000 cells for each group were analyzed. **b** A total of 12 different clusters representing nearly all cells from the TME were generated by Seurat package in R-studio and positioned with U-MAP representation. **c** Shown is a DotPlot listing the genes which the expression was used to identify the cell clusters generated in panel (**b**). **d** Shown is the proportion of individual cell populations from all cells analyzed. **e** UpsetPlot showing the interaction size (number of DEGs) after samples comparison in monocytes, macrophages and CD8 T cells. **f** Of the 63 genes in the monocyte cluster which the expression was restored in ABT treated mice, 13 were selected following their association with immunosuppression/immunotherapy resistance. Their individual expression are shown in RidgePlot to demonstrate the overall immunosuppressive phenotype of these monocyte cells. The global expression of the overexpressed genes (without Atf3) constitutes the TBI-Monocyte-Signature and is represented with a ViolinPlot. **g** The TBI-Monocyte-Signature identified in panel F is quantified in whole cells using FeaturePlot to highlight their expression within the monocyte cluster as defined in panel (**b**). **h** Single cell data sets generated by Zhang et al.[30] in B16F10 tumors collected from aged (20–22 months old) and young (6–8 weeks old) mice were used to analyze the expression of the TBI-Monocyte-Signature in monocytes as show by violin plots. Source data are provided as a Source Data file.

pH=6, 5 mM potassium ferrocyanide, 5 mM potassium ferricyanide, 150 mM Sodium chloride, 2 mM of magnesium chloride) for 4 h. Stained tumors were washed with PBS and a picture was taken with a stereo microscope (Leika M205FA).

## Single-cell analysis

For the analysis of the TME, tumors were dissociated and mPlum+ tumor cells sorted out by FACS. For the analyses of splenocytes and BM cells, cells were dissociated as indicated above and live cells sorted by FACS. Sorted cells ($1 \times 10^6$) were stained using CellPlex reagents according to the 10X genomics protocol. After labeling, cells were pooled and processed on the Chromium Controller according to manufacturer's protocol using the GEX 3' v3 kit aiming at a capture of 10,000 cells per sample. Sequencing libraries were generated as per manufacturer's instructions and sequenced on a NovaSeq S6000 aiming at 20,000 reads per cell for the gene expression libraries and 5000 reads per cells for the cell multiplexing library. Fastq files were aligned to the genome and reads counted using CellRanger version 6. Count matrices were analyzed using Seurat v4 (https://satijalab.org/seurat/index.html). CellPlex reads were normalized using CLR method. Mutual exclusivity of cell multiplexing label was visually confirmed. Gene expression reads were log-normalized with a scaling factor of 10,000, principal component analysis was performed using the 2000 most variable genes and data was embedded in two dimensions using Uniform Manifold Approximation and Projection computed on the first 15 principal components. Principal components used for analysis was defined by using 'ElbowPlot'. Cells with a fraction of mitochondria reads greater than two standard deviations above the mean were removed. Cells were clustered based on a nearest-neighbor graph construction using Seurat V4 with default parameters and resolution parameter at 0.4. Clusters were annotated by inspecting the expression of the differentially expressed genes with a log2 fold change of at least 0.15, a difference in proportion of positive cells of at least 0.01 and an adjusted $p$-value of less than 0.05 based on a wilcoxon rank sum test. We also inspected the expression of classically reported markers in the literature.

## Differentially expressed genes identification and gene set enrichment analysis

Gene set enrichment analysis (GSEA) analysis were performed on the DEG identified as above using the fgsea package (Release 3.17). The Biological process:GO were tested after downloading pathways from the MSIgb database (BP: subset of GO, https://www.gseamsigdb.org/gsea/msigdb/mouse/genesets.jsp?collection=GO:BP). Only the pathways with an adjusted $p$.value adj <0.05 were considered as statistically different.

## Spheroid infiltration assay

Spheroids were formed following plating $1 \times 10^4$ MC38 cells in a 96-well round bottom ultra-low attachment cell plate (LSBio, PrimeSurface® 3D Culture Spheroid plates: Ultra-low Attachment (ULA) Plates). 3 days after, when spheroids were formed, $1 \times 10^5$ freshly isolated splenocytes were added and coculture for 5 days after which spheroids were washed 3 times in PBS and enzymatically dissociated in trypsin 0.25% (Multicell) coupled with frequent pipetting during 5–10 min. Dissociated spheroid cells were analyzed by flow cytometry.

## Bioluminescence assay

To measure in vivo bioluminescence, p16-3MR mice were anesthetized using isoflurane and injected intraperitoneally with water-soluble coelenterazine (CTZ; NanoLight Technology™) at a concentration of 1 mg/mL in 0.9% NaCl (Sigma). 12 min post-injection, mice were imaged using the Epi-Fluorescence & Trans-Fluorescence Imaging System (Labeo Technologies). For the imaging of resected tissues, spleens were directly soaked in a 0.1 mg/mL diluted coelenterazine solution (in PBS), and immediately analyzed.

## Gene expression by RT-qPCR

RNA from splenocytes and monocytes was extracted with the RNeasy® Mini (Qiagen) after tissue and cell disruption with a rotor-stator. A total of 1 μg of extracted RNA was reverse-transcribed into cDNA using the QuantiTect Reverse Transcription Kit (Qiagen). Retrotranscript cDNA was diluted 1/5 with RNAase-free water. The real-time PCR reaction was performed on a light cycler system (Roche) in 96 wells using the SYBR-Green reagent (PowerUP Thermofisher) and specific primers in a final volume of 20 μL. Gene transcripts were normalized to *Gapdh* and *B2mg*. Relative mRNA compared to the control group was calculated using the comparative cycle threshold (CT) method (2− ΔΔCt). The following primers used for PCR:

    Mousse p21 – Forward – TTGTCGCTGTCTTGCACTCTGGT
    Mousse p21 – Reverse – AGACCAATCTGCGCTTGGAGTGAT
    Mouse Gapdh – Forward – GAAGGTCGGTGTGAACGGA
    Mouse Gapdh – Reverse – GTTAGTGGGGTCTCGCTCCT
    Mouse B2m – Forward – CTGCAGAGTTAAGCATGCCAGTA
    Mouse B2m – Reverse – TCACATGTCTCGATCCCAGTAGA
    Mouse p16 – Forward – CAGGGCCGTGTGCATGA
    Mouse p16 – Reverse – CATCATCACCTGAATCGGGGT

## CD8 T cell proliferation in vitro assay

Tumors were excised on day 22 after inoculation and dissociated with the mouse tumor dissociation kit (Miltenyi). All cells were stained with 1 μM of CFSE (Celtrace-Thermo fisher) for 20 min before adding complete media containing 10% FBS to absorb any unbound dye. Then, cells were plated in a 96-well plate ($1.5 \times 10^5$ cells/well) in complete RPMI 1640 supplemented with 0.1% βmercapto-ethanol, 2 mM L-glutamine and 30 UI.mL of IL-2. To activate T cells, CD3/CD28 beads (Dynabeads, Thermofisher) were added at a ratio of 2 beads per cell (2:1). The proliferative capacity of T cells was evaluated 72 h after activation by flow cytometry using the zombie-NIR live staining (Biolegend), CD3-AF700, and CD8-BV421 antibodies (Biolegend). The proportion of non-proliferative CD8 T cells was evaluated using the FlowJo v10 software. Were indicated, CD11b+ cells were removed using the PE-Selective dissociation kit for mouse cells (StemCell) following the manufacturer steps and using 3 μg of CD11b PE-coupled antibody

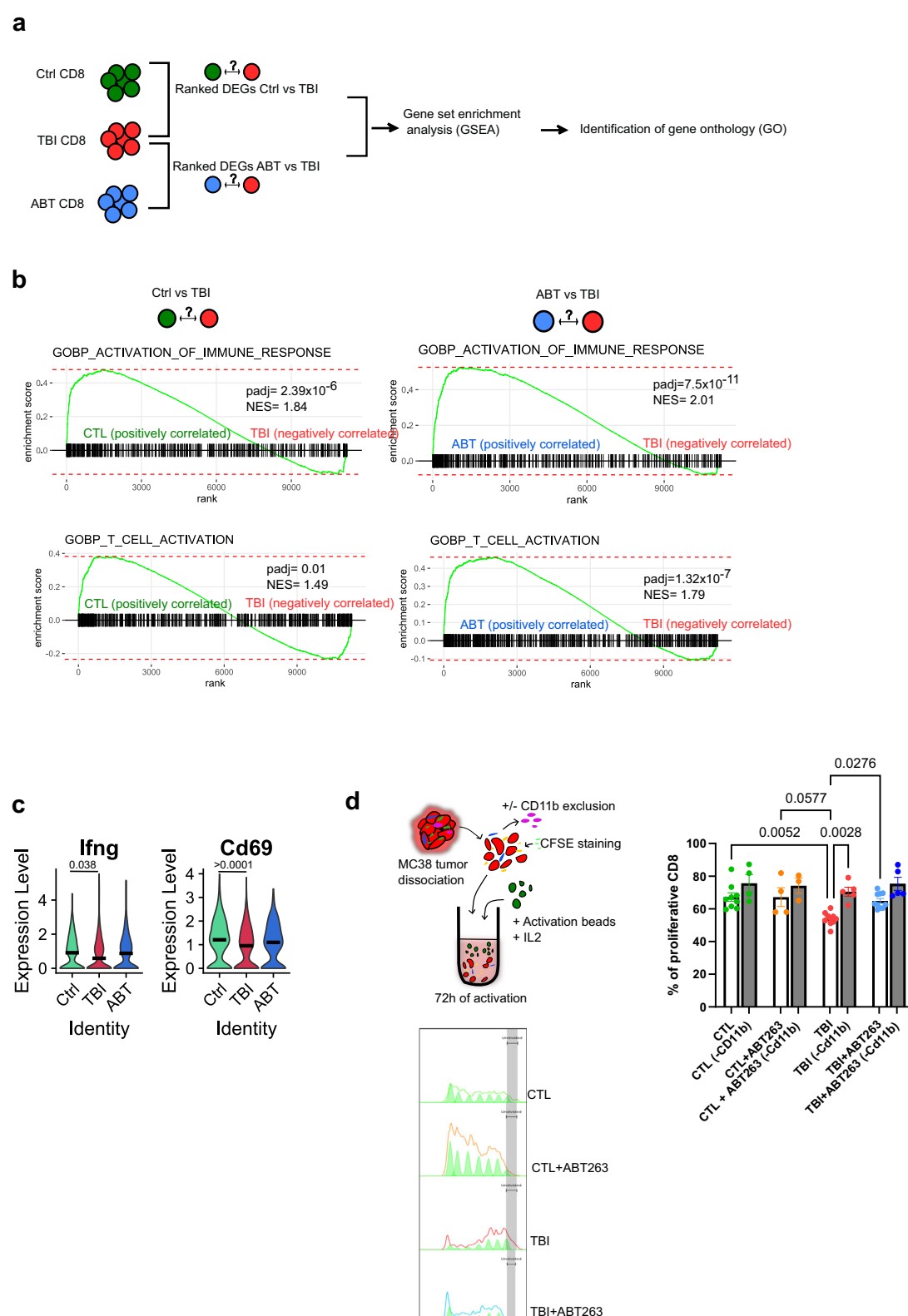

(Biolegend). A fraction of myeloid cells were used for qPCR analysis after lysing in RLT (Qiagen).

**Splenic CD8 T cell activation by APC**
Splenocytes were plated in a 96-well plate ($4 \times 10^5$ cells/well) in RPMI 1640 supplemented with 0.1% βmercapto-ethanol, 2mM L-glutamine and 30UI.mL of IL-2. Cells were then loaded with the gp33 peptide (Iba-

lifesciences) at a concentration of $1 \mu M$ for $12 h$ with $50 ng mL$ of LPS (Sigma). Fresh transgenic T cells specifically recognizing the gp33 peptide were purified from the spleen of P14 mice (Jackson, P14 TCRVα2Vβ8) using the EasySep Mouse T cell isolation kit (Stem cells). Purified T cells were then stained with $1 \mu M$ of CFSE (Celtrace-Thermo fisher) and cultivated with gp33 loaded cells at a concentration of $1 \times 10^5 T$ cells per well. After $72 h$, T cells proliferation was analyzed by

**Fig. 5 | Impaired activation and proliferation of CD8 T is mediated by monocytes and partially rescued by ABT263. a** GSEA of CD8 T cells from our RNA transcriptomic data using the fgsea package to identify enriched GO. $n = 1421$ CD8 T cells in CTL (from three different tumors), $n = 1073$ CD8 T cells in TBI (from 3 different tumors), $n = 1072$ CD8 T cells in ABT (from three different tumors). **b** Shown are the EnrichmentPlot with statistical information (*p*-value adjusted or padj, and the Normalized Enrichment Score or NES) from CD8 T cells GSEA for the selected specific GO: T cell activation and Activation of immune response. **c** Violin plots showing the transcriptomic expression of Ifny and Cd69 effector markers in CD8 T cells. FindMarkers function with Wilcoxon Rank Sum test. $n = 1421$ CD8

T cells in CTL (from three different tumors), $n = 1073$ CD8 T cells in TBI (from 3 different tumors), $n = 1072$ CD8 T cells in ABT (from three different tumors). **d** Schematic representing the ex vivo proliferative capacity of CFSE stained CD8 T cells within the TME, with or without myeloid cells (CD11b+), from the indicated groups following stimulation with activation beads. The percentage of proliferative CD8 T cells was evaluated using the FlowJo software and proliferation tools. $n = 9$ CTL, $n = 4$ CTL (-CD11b), $n = 4$ CTL + ABT263, $n = 4$ CTL + ABT263 (-CD11b), $n = 11$ TBI, $n = 5$ TBI (-CD11b), $n = 10$ TBI + ABT263, $n = 5$ TBI + ABT263 (-CD11b). Ordinary one-way ANOVA with Tukey correction. Shown is the mean ± SEM. Source data are provided as a Source Data file.

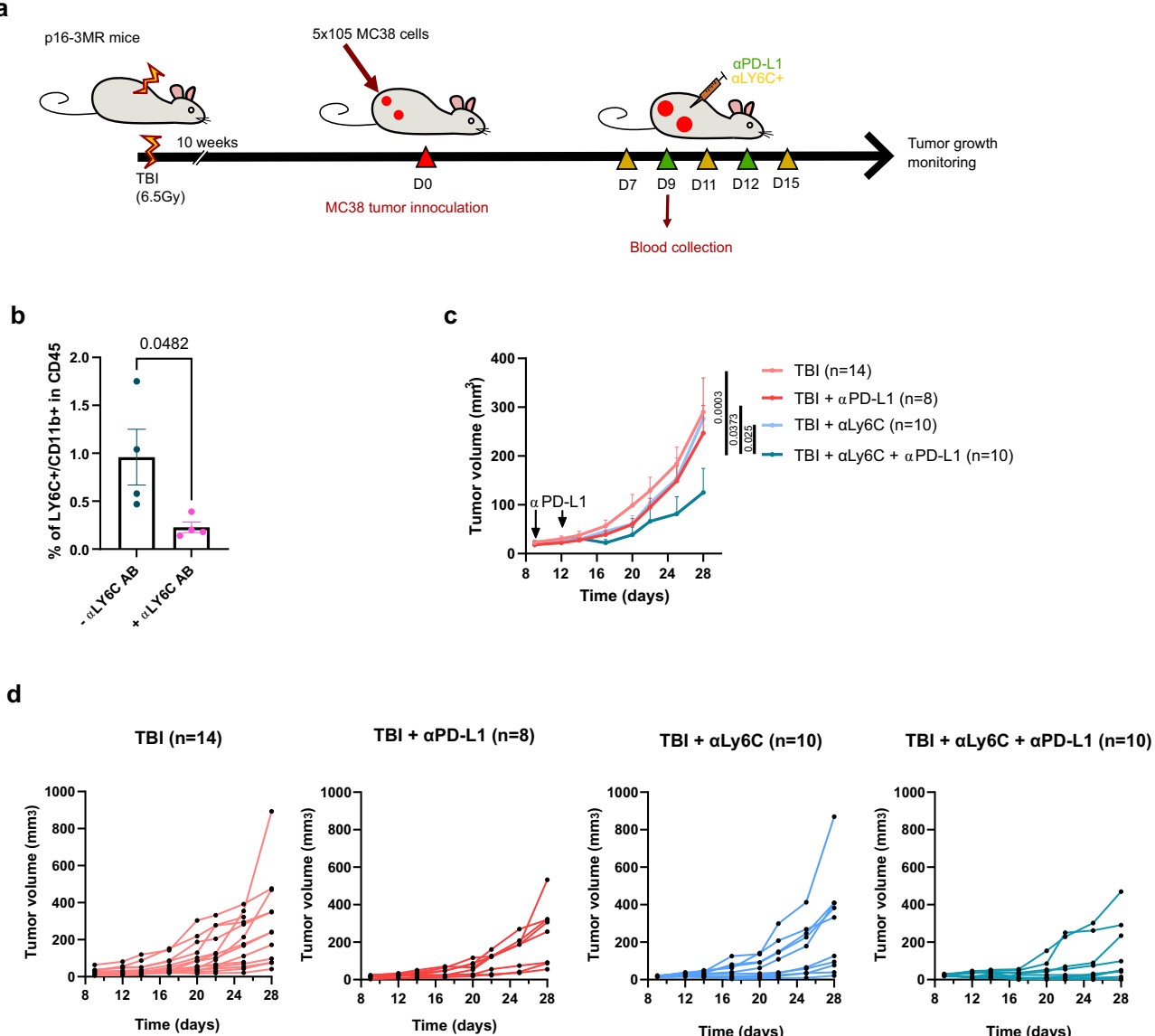

**Fig. 6 | Depletion of Ly6c+ myeloid cells partially overcomes resistance to αPD-L1. a** Schematic of the experiment. In brief, p16-3MR mice were injected on each flank with $5 \times 10^5$ MC38 tumor cells expressing the mPlum fluorescent protein. Mice received a αLy6c blocking antibody by intraperitoneal injection on days 7, 11, and 15 and/or a αPD-L1 blocking antibody on days 9 and 12. **b** Shown is the proportion in blood of CD11b + /Ly6c+ cells in the CD45+ population as measured by flow-cytometry two days after the first injection of αLy6c antibodies (day 9 post tumor inoculation). $n = 4$ independent animals. Two unpaired *T*-test. Shown is the

mean ± SEM. **c** Tumor growth was evaluated for each of the indicated groups of mice. Each line represents the mean tumor growth over 28 days or until the mouse had to be removed from the study. The total number of tumors per group is indicated in parenthesis. 2way ANOVA or Mixed-effect analysis with Tukey correction. Shown is the mean ± SEM. TBI group is the same as in Fig.1d and shown here again to facilitate comparison. **d** Shown is the size of each individual tumors at the indicated timepoint. The total number of tumors per group is indicated in parenthesis. Source data are provided as a Source Data file.

flow cytometry using CD3 (CD3-AF700, Biolegend), CD8 (CD8-BV421, Biolegend), and a viability dye (NIR-Zombie, Biolegend). FlowJo V10 software is used to analyze cell proliferation.

## OVA-DQ phagocytosis and processing assay

Splenocytes were plated in a 96-well plate ($1 \times 10^6$ cells/well) and OVA-DQ (DQ Ovalbumin, Thermofisher) was added at a concentration of 100 μg/mL for 30 min to allow its phagocytosis by APC. OVA-DQ was then washed and its processing by macrophages (F4/80+, CD45+) and dendritic cells (CD45+, CD11c ) analyzed 3 h later by flow cytometry (Digested fragment have BODIPY dye). The following antibodies were used: CD11c-APC (Biolegend), F4/80-APC/Cy7 (Biolegend), CD45-PE-Cy7 (Biolegend), 7AAD (Biolegend).

## Statistical analysis

Graph-pad Prism version 9 was used for the statistical analysis and graphical representations. All data showed are the mean +/− SEM. A one-way ANOVA (equal standard deviations, SDs) or Brown-Forsythe (unequal SDs) assessed the statistical significance for RT-qPCR, tumor immune infiltration, and T cell proliferation. For tumor growth, we used a two-way ANOVA or the mixed-effect analysis when the group sizes were unequal. We use the Tukey tests for the multiple comparison tests following Prism's recommendations. SDs differences were calculated with Brown-Forsythe and Bartlett's test. Normality and gaussian distributions of data set were measured with D'Agostino & Pearson test and Shapiro−Wilk test.

## Reporting summary

Further information on research design is available in the Nature Portfolio Reporting Summary linked to this article.

# Data availability

The single-cell transcriptomic data generated in this study have been deposited in the GEO database under accession code GSE256486. Data used in Fig. 5g, h from Zhang C, Lei L, Yang X, et al. (ref. 30. in the manuscript) are available in the scRNA database under the repository name SCP1261. The remaining data are available within the Article, Supplementary Information or Source Data file. Source data are provided with this paper.

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

## Acknowledgements

The authors acknowledge the service of the CHUSJ platforms and animal facility for their help on this project. This work was supported by a grant from the Canadian Institute of Health Research (PJT-169017) to C.B. and by the Cancer Research Society (CRP-159385) to C.B. We also would like to acknowledge the support of the Fondation Charles Bruneau for their support in the cost of using flow cytometry and other specialized platforms. D.M. is supported by a fellowship from le Fonds de recherche Santé Québec.

## Author contributions

D.M. performed the experiments; D.M., V.L., G.M.B., and S.L. processed samples for single-cell experiment and analyzed the associated data; O.L. generated and validated mice; V-P.L and H.D., designed and supervised individual experiments; C.B. designed and supervised the study; D.M and C.B. wrote the manuscript with the contributions from all the authors.

## Competing interests

The authors declare no competing interests
