## [Peer Review File · Nature Communications]

Senescence drives immunotherapy resistance by inducing an immunosuppressive tumor microenvironmentREVIEWERS' COMMENTS:

Reviewer #1 (Remarks to the Author): with expertise in senescence, cancer

The manuscript entitled “Therapy-induced senescence drives immunotherapy resistance by inducing an immunosuppressive tumor microenvironment” by Damien Maggiorani and collaborators claims that senescence induced by anticancer treatments results in the generation of an immunosuppressive microenvironment that induces the failure of immunotherapy in the control tumor growth.

For this, the authors rely on an experimental system that basically consists in the subcutaneous injection of a tumor cell line in mice previously exposed to cell senescence-inducing treatments (irradiation or doxorubicin). This is not a faithful representation of what the authors intend to demonstrate, that senescence-inducing therapies result in immunotherapy resistance since treatments are not applied to mice bearing tumors. Tumor cell senescence after therapy would be better modelled by applying senescence-inducing treatment after tumor growth.

Another serious consideration is the total lack of experimental evidence showing that the in vivo systems are actually behaving as they expect. There is no proof of senescence induction after treatments, there is almost no proof of senescent cell removal in the genetic ablation of senescent cells model.

A key issue is that senescence induction is claimed to produce immunosuppression that limits immunotherapy effectiveness. For this, authors use anti-PD-L1 to control tumor growth which is successfully achieved only in control mice but not in those that were treated to induce senescence prior to tumor cell injection. Removal of the senescent cells using a genetic system or a senolytic drug restores immunotherapy and tumor growth control. However, tumor growth in animals treated to induce senescence is seriously compromised and this is not restored by removing senescent cells. The authors claim that this might be mediated by “a modulation of the immunosuppressive TME composition” caused by senescence, but this is difficult to reconcile given the failure to restore robust tumor growth after removal of senescent cells. If we follow the author’s thinking and tumor growth is delayed by senescence modulating an immunosuppressive microenvironment, senescence removal should have a clear impact on tumor growth and not only on restoring immunotherapy effectiveness.

This limitation of immunotherapy after senescence induction is claimed to be produced by monocytes limiting the activation and proliferation of CD8 T cells. To support this, the authors looked at immune cell infiltration in tumors and gene expression changes related with immune modulation. However, there are no candidate factors identified as mediating these effects and thus we are left with an interesting description that is not further substantiated by offering potential hits to revert this immunosuppressive tumor microenvironment.

Reviewer #2 (Remarks to the Author): with expertise in senescence, cancer immunology

Maggiarani et al. provided a very interesting study showing that therapy-induced senescence reprograms suppressive tumor microenvironment, leading to resistance to tumor immunotherapy. They further performed in vivo experiments that the elimination of senescent cells via genetic regulation and senolytic drug can restore immunotherapy efficacy. In addition, they identified that increased monocytes with an immunosuppressive phenotype in the senescent models could be responsible for immune suppression and immunotherapy resistance.

Overall, this manuscript provided risible experiments and an excellent research design, and all the studies supported the claims and conclusions. The topic is novel and clinically significant. However, several significant concerns/suggestions exist in this study that need to be addressed to improve this manuscript.

1) In this manuscript, the authors utilized the two therapy-induced senescence models for the studies. However, the authors did not characterize the senescence development in these models. Furthermore, the information for the senescent populations in these two therapy-induced models and senescent cell elimination studies is unclear, especially immune cell components.

2) The causative relationship between the therapy-induced senescence and the increased population of monocytes is unclear. Are the monocytes senescent monocytes in the two models?

3) In addition, whether the increased suppressive monocytes are responsible for the immunotherapy resistance is unclear. The molecular processes for the therapy-induced increase in the monocytes are unknown.

4) It will be essential to determine whether the deletion/decrease of this ly6c+ myeloid population will enhance or break the immunotherapy resistance for tumor immunotherapy.

Reviewer #3 (Remarks to the Author): with expertise in cancer immunotherapy, radiotherapy

The Manuscript by Maggiorani and authors is an addresses the clinical concern of immunotherapy use in the face of SOC therapy. Use of IO in the clinic has been fairly established but work exploring resistance to these therapies are imperative. The authors hypothesize that therapy induced senescence drives IO resistance and propose a novel mechanism through how use of SOC can enhance the immunosuppressive profile of infiltrating myeloids, diminishing the anti-tumoral response seen in the face of aPDL1. Strengths of this work include utilizing different models of TIS including use of RT in the form of total body irradiation (TBI) and Doxorubicin that have been previously established to study this phenotype. Likewise, the authors use single-cell transcriptomics to address a potential mechanism in the TME for diminishing effects of IO. These data sets are valuable to the advancement of studying senescence, as a scientific phenomenon pertaining to aging and immune surveillance and response.

However, there are several major concerns with the work. How the models chosen, especially TBI, pertains to cancer is unclear. TBI is lymphodepleting, almost never used for solid tumors, especially not in combination with immunotherapy. It is unclear if the authors are just recapitulating a state of recurrent metastatic disease, often with large burden of disease, having been through several lines of therapy, with persistent lymphopenia, and then given immunotherapy with little to no effect. Undoubtedly senescence would logically play a role in that setting. The challenge is in using TBI with so many off target effects on numerous cell types, followed by subcutaneous tumor implantation. Other concerns include number of mice used in groups, the magnitude of the effect, data presentation, and general

appearance and understanding of graphs. Similarly, while the authors descriptively characterize the immune populations in the TME, there is a lack of functional depletion studies that should be properly evaluated.

Major

1. The use of TBI is not translational and does not recapitulate any “therapy” in solid tumors. If the authors are using it as a model of senescence, it should be stated and avoid the label therapy induced. More concerning is that TBI affects all lymphatics, the entirety of the bone marrow, the GI tract among other things. The persistent lymphodepleting effects resulting from that are huge threat to the validity of the findings particularly as it pertains to immunotherapy. Independent of senescence, how it is expected the immunotherapy would work after TBI? Can the authors verify that the marrow has recovered.
2. While TBI is being used to induced senescence here, although many other processes are induced with total body irradiation, it is highly misleading to call this “therapy” induced senescence. This is not a therapy that is ever utilized for solid tumors. Even overlooking the translational weakness of this model, mechanistically it is challenging to decipher what other biological processes this therapy induced.
3. The choice of subcutaneous models is not faithful to the biology of any disease outside perhaps skin cancer.
4. Figure 1 is vague. Unclear why the authors split the curves into 3 subfigures in D if the focus is on comparing what happens to response to IO response in the context of TIS. There appears to be no survival difference between any of the groups as the median survival is approximately the same (~40 days). The tumor volume spider plots is hard to extract much from as it only goes till day 28. But it appears from the first rate that TIS reduces take rate of tumors, more so than response of the tumor to a-PDL1.
5. Validation of senescence is not present in the authors in vivo models despite use of P16-3MR mice (Figure 1 and 2). Can the authors contextualize P16ink4a expression within circulating lymphocytes following TIS to validate?
6. In figures 1 & 2, Two tumors are implanted per mouse simultaneously? While this saves

on the number of mice, it makes it challenging to monitor tumor growth and assess survival if one tumor grows before the other.. In Figure 2 yet again, the TBI appears to be decreasing take rate. Without validation of senescence, it is possible that the GCV itself especially along with aPD1 is acting as a therapeutic modality. Particularly since the volume of the tumor was so small at presentation.

7. The drug ABT263 has pleiotropic effects. It is unclear if other processes are not involved. Can the authors repeat the experiments in the p16-3MR

Mice and show absence of an effect there to show dependency on this biological phenomena?

8. In Figure 4, the authors present an abscopal effect model using cell lines that share the same antigen and assume tumor homogeneity. The effects with ABT263 could be related to enhanced perfusion and improved drug delivery to tumor and nothing to do with senescence. In these models death from distant tumor versus death from primary tumor should be plotted separately to evaluate the effect

9. The scRNA seq is to be commended. However, these are mouse tumors in flank models and the generalizability is limited. The authors describe the APC capacity to present antigen from the splenocytes as not being affected. Have the authors considered intratumoral APC functionality?

10. The paragraph explaining the “profile of monocytes in the spleen and bone marrow” seems to not tie in well on its own. Can the authors find a way to better tie this in to earlier figures such as Figure 1 to establish their reasoning for pre-treating with TBI.

11. In Figure 6C the authors demonstrate CD8+ functionality through IFN γ and CD69+ expression. Are these differences statistically significant?

12. In Figure 6D the authors separate “with CD11b and without CD11b” into two graphs. Can the authors consolidate these two graphs to better compare. It appears there is very little difference between the presence of APCs or without APCs.

13. Have the authors considered Gain of function/Loss of function experiments utilizing

myeloid depleting strategies? In the discussion it is claimed that the effect of TIS is due to immunosuppressive myeloid cells.

Minor

- 1.) Figure With Induced senescent in TBI is underpowered compared to control in Figure 1B
- 2.) Can the authors elaborate on what the control is in Figure 2B-D in figure legend. I assume it is no TBI but please clarify.
- 3.) In figure 3D the n size for each group are disparate. Can the authors provide spider plots or individual mouse tumor sizes to account for overpowering.
- 4.) In Figure 3F the plots are described as “% of cells” in the TME. Can the authors elaborate on what this means. Is it a % of total cells, CD45+, Live?
- 5.) Can the authors provide individual mouse tumor sizes in Figure 4B to account for disparate group sizes.

Reviewer #4 (Remarks to the Author):

‘I co-reviewed this manuscript with one of the reviewers who provided the listed reports. This is part of the Nature Communications initiative to facilitate training in peer review and to provide appropriate recognition for Early Career Researchers who co-review manuscripts.’

Rebuttal – manuscript NCOMMS-23-33111

Reviewer #1

Comment #1

The authors rely on an experimental system that basically consists in the subcutaneous injection of a tumor cell line in mice previously exposed to cell senescence-inducing treatments (irradiation or doxorubicin). This is not a faithful representation of what the authors intend to demonstrate, that senescence-inducing therapies result in immunotherapy resistance since treatments are not applied to mice bearing tumors. Tumor cell senescence after therapy would be better modelled by applying senescence-inducing treatment after tumor growth.

Response: We agree with the reviewer that such an experiment would complement our manuscript. As such, we have performed the requested experiment by irradiating locally MC38 tumors (12Gy single dose). Induction of senescence was confirmed in these tumors and 2 weeks after mice were treated with a blocking α PDL-1 antibody (new Fig. S3). This protocol which is equivalent to therapy induced senescence (TIS) showed that α PDL-1 therapy is also compromised in this context.

However, exposure to systemic chemo (Doxo) or TBI is highly relevant in the context of ICI treatments as most if not all cancer patients that are treated with ICI have an impaired immune system either because of age or because of previous rounds of treatments or both. To avoid any confusion, in the revised manuscript we only use the term TIS to describe the new results presented in Fig. S3.

Comment #2

Another serious consideration is the total lack of experimental evidence showing that the in vivo systems are actually behaving as they expect. There is no proof of senescence induction after treatments, there is almost no proof of senescent cell removal in the genetic ablation of senescent cells model.

Response: We indeed used total body irradiation (TBI and Doxo) to induce senescence and as a premature aging model like we and others did in the past (Le et al. Aging Cell 2010, Palacio et al. Aging Cell 2019, reviewed in Wang et al. Nature Reviews Cancer 2022). That said, we agree we should have included experimental data showing the expression of senescence markers. In the revised manuscript we added additional results showing:

A) Increased p21/p16 mRNA in the spleen 12 and 4 weeks after exposure to TBI and Doxo treatments respectively (Fig 1B). Note that p16 was not increased by Doxo, a phenotype also observed by others (Postmus, Kruit et al. 2023).

B) Increased luminescent signal in the spleens of p16 3MR mice 12 weeks after their exposure to TBI and decreased luminescence after the injection of ganciclovir (Fig. 1C).

Comment #3

Tumor growth in animals treated to induce senescence is seriously compromised and this is not restored by removing senescent cells. The authors claim that this might be mediated by “a modulation of the immunosuppressive TME composition” caused by senescence, but this is difficult to reconcile given the failure to restore robust tumor growth after removal of senescent cells. If we follow the author’s thinking and tumor growth is delayed by senescence modulating an immunosuppressive microenvironment, senescence removal should have a clear impact on tumor growth and not only on restoring immunotherapy effectiveness.

Response: We also were surprised that removal of senescent cells did not accelerate tumor growth in TBI/Doxo treated mice. We have rephrased this sentence to make it clear that removal of senescent is insufficient to reinstate normal tumor growth (page 6 1st paragraph). Reasons for this are unknown but one may speculate that senescence induces irreversible changes. Slower tumor growth has also been observed in aged mice (ref 29 and 30 in the manuscript). In any case, slower tumor growth in TBI mice should have favored tumor elimination by immune cells which is the opposite of what we observed.

Comment #4

This limitation of immunotherapy after senescence induction is claimed to be produced by monocytes limiting the activation and proliferation of CD8 T cells. To support this, the authors looked at immune cell infiltration in tumors and gene expression changes related with immune modulation. However, there are no candidate factors identified as mediating these effects and thus we are left with an interesting description that is not further substantiated by offering potential hits to revert this immunosuppressive tumor microenvironment.

Response: We believe the resistance arise from multiple SASP factors and is unlikely to be mediated by one factor. We have added a sentence in the discussion on page 14 to reflect this possibility. Instead, we provide data showing that the phenotype is reversible using either senolytic drugs or following the elimination of Ly6c+ monocytes (new figure 6). How monocytes are primed in the context of senescence is a complex question that needs to be address separately.

Reviewer #2

Comment #1

In this manuscript, the authors utilized the two therapy-induced senescence models for the studies. However, the authors did not characterize the senescence development in these models Furthermore, the information for the senescent populations in these two therapy-induced models and senescent cell elimination studies is unclear, especially immune cell components.

Response: Yes we agree, these results were added. See response to reviewer 1 comment #2

Comment #2

The causative relationship between the therapy-induced senescence and the increased population of monocytes is unclear. Are the monocytes senescent monocytes in the two models?

Response: Based on our transcriptomic single-cell analysis, monocytes do not express classical senescent markers (p16, p21, or related SASP factors). Instead we believe monocytes are mostly primed as immunosuppressant by systemic SASP factors. We added a sentence on page 11 (last sentence of the 1st paragraph) that indicates this.

Comment #3

In addition, whether the increased suppressive monocytes are responsible for the immunotherapy resistance is unclear. The molecular processes for the therapy-induced increase in the monocytes are unknown.

Response: We actually do not see a significant increase in the number of monocytes. Instead we see a decrease in the ratio of CD8/CD11b cells in tumors from TBI mice and show that these monocytes are more immunosuppressive. How at the molecular level these monocytes become more immunosuppressive is a complex question that needs to be address separately.

Comment #4

It will be essential to determine whether the deletion/decrease of this Ly6c+ myeloid population will enhance or break the immunotherapy resistance for tumor immunotherapy..

Response: We agree and we now provide new results showing that the elimination of Ly6c+ monocytes from tumors of TBI mice partially restores sensitivity to immunotherapy (new Fig. 6).

Reviewer #3

Comment #1

The use of TBI is not translational and does not recapitulate any "therapy" in solid tumors. If the authors are using it as a model of senescence, it should be stated and avoid the label therapy induced. More concerning is that TBI affects all lymphatics, the entirety of the bone marrow, the GI tract among other things. The persistent lymphodepleting effects resulting from that are huge threat to the validity of the findings particularly as it pertains to immunotherapy. Independent of senescence, how it is expected the immunotherapy would work after TBI? Can the authors verify that the marrow has recovered.

Response: We fully agree with the reviewer and we indeed used TBI/Doxo as a model of senescence. We now reserved the term therapy induced senescence (TIS) only to the new data showing local tumor radiotherapy (new Fig. S3). That said, we also added data showing complete immune cells recovery in blood 12 or 4 weeks post TBI or Doxo respectively (new Fig. S1).

Comment #2

While TBI is being used to induced senescence here, although many other processes are induced with total body irradiation, it is highly misleading to call this "therapy" induced senescence. This is not a therapy that is ever utilized for solid tumors. Even overlooking the translational weakness of this model, mechanistically it is challenging to decipher what other biological processes this therapy induced.

Response: We agree and as stated above we now limit the use of TIS to our new data showing that local radiotherapy also impairs immunotherapy. However it is important to mention that we have used two distinct models (TBI and Doxo) and thus we are confident that the phenotype is not restricted to the effects of TBI.

Comment #3

The choice of subcutaneous models is not faithful to the biology of any disease outside perhaps skin cancer.

Response: We agree with the reviewer but we are not aware of a cancer cell line that respond to immunotherapy and that grows orthotopically. The intracecal injection of MC38 cells is possible but technically challenging with the result that their engraftment rate was shown to be only 67% with sometimes very large or very small tumors (PMID: 36225595). The only other cancer cell line we know that responds, at least partially, to immunotherapy is the B16 melanoma cell line which is of no use as it would still be subcutaneous. We tried injecting the MC38 cell line iv as a model of metastasis but the experiment failed as more metastasis were formed in the TBI group (both in number and size of nodules – see data below). Such differences between the control and TBI group prevented us to measure the effect of ICI as

the efficacy is tightly dependent on tumor size at the time of treatment. Increased metastasis formation in a senescent environment has been reported before (PMID: 27979832).

Comment #4

Figure 1 is vague. Unclear why the authors split the curves into 3 subfigures in D if the focus is on comparing what happens to response to IO response in the context of TIS. There appears to be no survival difference between any of the groups as the median survival is approximately the same (~40 days). The tumor volume spider plots is hard to extract much from as it only goes till day 28. But it appears from the first rate that TIS reduces take rate of tumors, more so than response of the tumor to a-PDL1.

Response: Growth curves were split into 3 subfigures to improve clarity as tumor growth in absence of PD-L1 treatments is not the same. In the revised manuscript figure 1 was merged with figure 2. Increased survival is observed in control mice receiving α PDL-1 and in TBI mice treated with GCV prior to receiving the α PD-L1. Doxo treated mice do not show increase survival as these mice do not upregulate p16 expression and therefore have no benefit from the GCV treatments that kills only p16 positive cells.

Comment #5

Validation of senescence is not present in the authors in vivo models despite use of P16-3MR mice (Figure 1 and 2). Can the authors contextualize P16ink4a expression within circulating lymphocytes following TIS to validate?

Response: We agree and have included these data in Fig. 1B and 1C). See response to comment #2 from reviewer 1.

We did not find increased p16 expression in lymphocytes based on our single-cell transcriptomic data. However, because of the low abundance of p16 transcripts it is not surprising that single-cell data did not show increase expression. Instead, we tried a different model, that is the p16 tandem-dimer Tomato (tdTom) mice reporter (Liu et al. PNAS 2019 – PMID: 30683717). Yet, this model also failed at showing increase p16 expression in lymphocytes isolated from the spleen (p16 was however significantly increased in macrophages and splenic stromal cells – see below). In brief, $p16^{tdTom}$ mice were exposed to TBI and 8 weeks later splenic cells were dissociated and p16 (tdTom) expression determined by flow cytometry. These results are shown for your information and are not part of the manuscript.

Comment #6

In figures 1 & 2, Two tumors are implanted per mouse simultaneously? While this saves on the number of mice, it makes it challenging to monitor tumor growth and assess survival if one tumor grows before the other.. In Figure 2 yet again, the TBI appears to be decreasing take rate. Without validation of senescence, it is possible that the GCV itself especially along with aPDL1 is acting as a therapeutic modality. Particularly since the volume of the tumor was so small at presentation.

Response: In the revised manuscript former figure 1 and 2 have been merged into a new figure 1. Yes, in most mice two tumors were implanted per mouse. Mice were removed from the study if asynchrony was observed in tumor growth. Finally, the last dose of ganciclovir (GCV) was administrated 5 days before tumor cells injection and thus is highly unlikely to affect tumor growth.

Comment #7

The drug ABT263 has pleiotropic effects. It is unclear if other processes are not involved. Can the authors repeat the experiments in the p16-3MR mice and show absence of an effect there to show dependency on this biological phenomena?

Response: In figure 1 we used p16-3MR mice and depleted senescent cells using GCV to show rescue of IR-induced resistance. ABT263 was also administered prior to tumor inoculation and is thus unlikely to have an effect on tumor growth.

Comment #8

In Figure 4, the authors present an abscopal effect model using cell lines that share the same antigen and assume tumor homogeneity. The effects with ABT263 could be related to enhanced perfusion and improved drug delivery to tumor and nothing to due =with senescence. In these models death from distant tumor versus death from primary tumor should be plotted separately to evaluate the effect

Response: We understand the reviewer's comment but this is not a concern in the current experimental setting. Indeed, the primary tumor (treated with 3 doses of RT) is always completely rejected so that the cause of death is only in response to the growth of the distant tumor. See graph of individual tumors in response to "minor" comment #5 below.

Comment #9

The scRNA seq is to be commended. However, these are mouse tumors in flank models and the generalizability is limited. The authors describe the APC capacity to present antigen from the splenocytes as not being affected. Have the authors considered intratumoral APC functionality?

Response: This is an excellent question for which we don't know the answer. We have frozen tumor homogenates (over 12-18 months old) so we don't know if the functions of APC would be fully preserved or if APC are sufficient in number in those homogenates. Nonetheless, we have added a comment in the revised manuscript to indicate that the functionality of APC in tumors may be impaired (page 9 last paragraph).

Comment #10

The paragraph explaining the "profile of monocytes in the spleen and bone marrow" seems to not tie in well on its own. Can the authors find a way to better tie this in to earlier figures such as Figure 1 to establish their reasoning for pre-treating with TBI.

Response: We agree with the reviewer and have removed this paragraph in the revised manuscript. The data is still presented in Fig. S8 and S9 and the data briefly mentioned in the discussion.

Comment #11

In Figure 6C the authors demonstrate CD8+ functionality through IFN γ and CD69+ expression. Are these differences statistically significant?

Response: These results (now Fig. 5C in the revised manuscript) are statistically different between control and TBI. The figure was annotated accordingly.

Comment #12

In Figure 6D the authors separate "with CD11b and without CD11b" into two graphs. Can the authors consolidate these two graphs to better compare. It appears there is very little difference between the presence of APCs or without APCs.

Response: We have combined the two graphs as recommended (now Fig. 5D).

Comment #13

Have the authors considered Gain of function/Loss of function experiments utilizing myeloid depleting strategies? In the discussion it is claimed that the effect of TIS is due to immunosuppressive myeloid cells.

Response: Yes we have performed this important experiment see response to comment #4 from reviewer 2.

Minor

1.) Figure With Induced senescent in TBI is underpowered compared to control in Figure 1B

Response: We agree and we have added mice in this group (n=6 to n=14 tumors).

2.) Can the authors elaborate on what the control is in Figure 2B-D in figure legend. I assume it is no TBI but please clarify.

Response: Yes the control group is without TBI or Doxo. The legend was modified accordingly.

3.) In figure 3D the n size for each group are disparate. Can the authors provide spider plots or individual mouse tumor sizes to account for overpowering.

Response: Yes we now present individual mouse tumor sizes (Fig. 2E in the revised manuscript).

4.) In Figure 3F the plots are described as “% of cells” in the TME. Can the authors elaborate on what this means. Is it a % of total cells, CD45+, Live?

Response: Indeed it is the % of cells in the live CD45+ fraction dissociated from tumors. The legend was modified to make it clear.

5.) Can the authors provide individual mouse tumor sizes in Figure 4B to account for disparate group sizes.

Response: Individual mouse tumor sizes are shown below but were not added to the revised manuscript.

Control mice

TBI mice

REVIEWERS' COMMENTS

Reviewer #2 (Remarks to the Author):

The authors have addressed my concerns, and the manuscript is significantly improved.

Reviewer #3 (Remarks to the Author):

The authors have adequately addressed my concerns

Reviewer #4 (Remarks to the Author):

I co-reviewed this manuscript with one of the reviewers who provided the listed reports.

This is part of the Nature Communications initiative to facilitate training in peer review and to provide appropriate recognition for Early Career Researchers who co-review manuscripts.